# Posterior Medial Meniscus Root Repair Using Two Transtibial Tunnels with Modified Mason–Allen Stitches: A Technical Note

**DOI:** 10.3390/medicina59050922

**Published:** 2023-05-11

**Authors:** Du-Han Kim, Ki-Cheor Bae, Chang-Jin Yon, Ji-Hoon Kim

**Affiliations:** Department of Orthopedic Surgery, Keimyung University Dongsan Hospital, Keimyung University School of Medicine, Daegu 42601, Republic of Korea

**Keywords:** meniscus, meniscus root tear, repair, arthroscopy, knee

## Abstract

Complete tear of the posterior medial meniscus root can result in a loss of hoop tension and increased contact pressure. Thus, medial meniscus posterior root tear (MMPRT) is increasingly recognized as an important pathology. Although several surgical techniques for MMPRT have recently been introduced, the ideal technique is not yet established. This technical note is aimed at introducing a novel surgical technique using two transtibial tunnels with modified Mason–Allen stitches in the treatment of MMPRT.

## 1. Introduction

A complete tear of the medial meniscus posterior root (MMPR) can cause a complete loss of hoop tension and increased contact pressure due to altered biomechanics of the normal knee joint, paralleling complete meniscectomy or radial tear [1,2]. As an aging society progresses, the incidence of medial meniscus posterior root tear (MMPRT) is increased. Furthermore, the high incidence in Asia (27.8%) has been attributed to squatting and sitting on the floor with folded legs [3].

No benefit in halting arthritic progression was obtained by use of nonoperative treatment or partial meniscectomy for the treatment of complete MMPRT [4,5]. Krych et al. reported the results of partial meniscectomy for MMPRT. They found that patients who undergo arthroscopic partial meniscectomy for MMPRTs still progress to significant arthritis and concluded that repair should be considered in select patients without degenerative changes [5]. Therefore, if possible, restoration of meniscal function by surgical repair is necessary.

Numerous surgical methods have recently been suggested for the repair of the MMPR. Among them, the popularity of a transtibial pullout technique has increased consequent to the idea that the contact pressure and contact are normalized [6]. Of the currently available stitches, the modified Mason–Allen (MA) stitch has been accepted as an effective technical approach [2,7].

In this study, we describe the procedure for the repair of MMPRT using two transtibial tunnels with modified MA stitches.

## 2. Methods and Results

### 2.1. Preoperative Evaluations and Indication

All included patients underwent plain radiography and MRI to confirm the MMPRT. We defined an MMPRT when two or more of the following signs appeared on MRI: the absence of an identifiable meniscus or a high signal that replaced the normal dark meniscus signal (i.e., the ghost sign) in the sagittal view, a vertical linear defect at the meniscus root in the coronal view, and/or a radial linear defect at the posterior insertion in the axial view [8].

The inclusion criteria were (1) an MMPRT on MRI in a patient with a Kellgren–Lawrence (K–L) grade 1 or less, (2) those willing to follow a rehabilitation process postoperatively, (3) a symmetric hip–knee–ankle angle less than 5°, (4) Outerbridge classification less than grade III, and (5) younger than 70 years of age.

### 2.2. Diagnostic Arthroscopy and Superficial Medial Collateral Ligament (sMCL) Release

General arthroscopic examination using anterolateral (AL) and anteromedial (AM) portals was performed according to routine. An arthroscope (ConMed Linvatec, Largo, FL, USA) was inserted through the AL portal, and working devices were inserted using the AM portal.

If MMPRT was confirmed, we performed sMCL release to provide ample working space. A ~3–4 cm vertical incision was made using a No. 15 blade at the anteromedial aspect of the proximal tibia (Figure 1). Then, we found the sMCL and sartorius fascia. To preserve deep MCL and proximal attachment of the sMCL, the sMCL was released more downward than the sartorius fascia using a periosteum elevator.

### 2.3. Preparation for Root Repair

An arthroscopic PassPort cannula (Arthrex, Naples, FL, USA) was inserted for the performance of a convenient procedure and to prevent twisting of the stitch. Landmarks relevant to the insertion of the MMPH, including tibial attachment of the posterior cruciate ligament, tibial medial eminence, and articular surface of the tibial plateau, were then identified. Unhealthy tissue removal of the torn meniscus edge was conducted using an arthroscopic shaver (ConMed Linvatec, Largo, FL, USA). For the creation of a bony bed, a curette was inserted through the AM portal, and bony preparation was performed (Figure 2).

### 2.4. MMPR Stitches

Passage of the Knee Scorpion suture passer (Arthrex, Naples, FL, USA) loaded with a No. 2 Ultrabraid (Smith and Nephew, Andover, MA, USA) through the AM portal was then performed. The separated segment of the medial meniscus posterior horn (MMPH) was penetrated using a Scorpion needle at about 5 mm medial point to a detached margin. The second stitch was penetrated in the anterior location of the first stitch, using the same method. The upper two strands of the stitches were pulled out and tied. Using the shuttle relay technique, the first stitch was exchanged with the second stitch to make a horizontal loop (Figure 3).

The Knee Scorpion suture passer (Arthrex, Naples, FL, USA) was reintroduced using the AM portal, and two vertical stitches penetrated just the medial side of the horizontal stitch.

### 2.5. Tibial Tunnel Making

Insertion of the Meniscus Root Repair System (Smith and Nephew, Andover, MA, USA) was conducted using the AM portal. The tip of the guide was placed in the most medial side of the decorticated site of MMPR. A 2.4 mm Kirschner wire (K-wire) was advanced through the guide system. The location of the K-wire was confirmed using an arthroscope via the AL portal. The second K-wire was placed parallel and about 5 mm laterally to the first tunnel (Figure 4). Once it was verified that the position of the K-wire was acceptable, the medial-side K-wire was removed first. A metal wire was inserted into the created tunnel, and then it was withdrawn through the AM portal using an arthroscopic grasper.

### 2.6. Repair of MMRT

The wire was pulled through the tibial tunnel. For the medial tunnel, two horizontal stitches and two inferior vertical stitches were passed, resulting in a total of four stitches. For the lateral tunnel, the two superior vertical stitches were passed. The sutures from both tunnels were tied over the anteromedial tibial cortex, with the knee at 30° flexion. An arthroscopic re-evaluation was conducted to check for repair of the torn posterior root and to restore tension within the entire medial meniscus (Figure 5).

## 3. Case Series

A total of 20 patients were included. Their mean age was 61.5 years (range: 55–68) and their mean preoperative alignment was 2.0° (range: 0.6–4.4°).

The International Knee Documentation Committee (IKDC) score and Lysholm score were compared preoperatively and 1 year postoperatively; preoperative and postoperative mean IKDC scores were 40.9 ± 11.8 and 69.5 ± 11.9, and preoperative and postoperative Lysholom scores were 60.2 ± 11.0 and 88.3 ± 9.99, respectively. Both clinical scores were significantly improved 1 year postoperatively compared with baseline. Upon image evaluation, there was no significant difference in HKA angle at 1 year after surgery (2.0° versus 2.8°, *p* = 0.08). Four of 20 knees had progressed osteoarthritis from KL grade 0 to grade 1.

## 4. Discussion

Findings from several clinical studies have shown an association between nonoperative treatment or meniscectomy with poor outcomes for the prevention or delay of osteoarthritis in patients with MMPRT [4,5]. Instead, repair of MMPRT can result in restoration of the hoop tension biomechanically. According to a systematic review and long-term clinical study, repair of the MMRT showed better outcomes and survivorship than other treatments for long-term follow-up [9]. Chung et al. found that root repair showed better results than meniscectomy in functional scores and survivorship for more than 10 years of follow-up. According to their study, the survival rates of the repair and meniscectomy groups were 79.6% and 44.4% at 10 years, respectively [9].

Among several techniques, repair of the MMPR using a transtibial pullout technique was shown to restore contact pressures to the normal states and allow for the dispersal of hoop stresses across the meniscus [10]. However, there is still considerable debate regarding the number of tunnels.

The goal of the two-tunnel technique is to recover the normal anatomy of the MMPHR attachments. However, some authors have expressed concern that a problematic tunnel coalition may lead to a narrow attachment area for the meniscus, which could cause breakage. The single-tunnel technique reduces operation time and avoids these disadvantages [2].

There are few clinical or biomechanical comparative studies with one or two tunnels. According to LaPrade et al., similar biomechanical results for the transtibial pullout technique were shown using one- and two-tunnel techniques. They conducted a human cadaveric study that involved a transtibial pullout repair using one or two tunnels, and they found no significant difference in ultimate failure loads between both techniques [6]. Instead, similar studies are being actively conducted in the field of rotator cuffs [11,12]. Park conducted a biomechanical comparative study of single-point and double-point repairs. They found that double-point fixation (modified double row) had significantly more footprint contact than single-row repair [11]. Quigley et al. also reported that double fixation was superior to single fixation biomechanically.

Therefore, we thought that the two-tunnel technique might provide greater advantages than the one-tunnel technique in biomechanical and biological healing. In addition, as a result of development of arthroscopic instruments such as the Meniscus Root Repair System (Smith and Nephew) and the Knee Scorpion suture passer (Arthrex), the operation time can be gradually reduced.

The MA stitch leads to minimal slippage and elongation of the longitudinally oriented fibers of the meniscus or tendon and provides a greater holding power [2]. In a comparative study of MMPRT repair using MA or simple stitches, the MA stitch showed significantly superior outcomes with respect to postoperative extrusion and root healing. [13]. By using these devices, there is also room for passing the thread through the tunnel, providing easier use of the double Mason–Allen stitches.

## 5. Conclusions

Although several repair techniques for MMPRT have recently been introduced, the ideal technique has not yet been established. Our technique might be beneficial in the restoration of the function of the medial meniscus in patients who present MMPRT. However, further clinical and biomechanical studies are necessary to validate this technique.

## Figures and Tables

**Figure 1 medicina-59-00922-f001:**
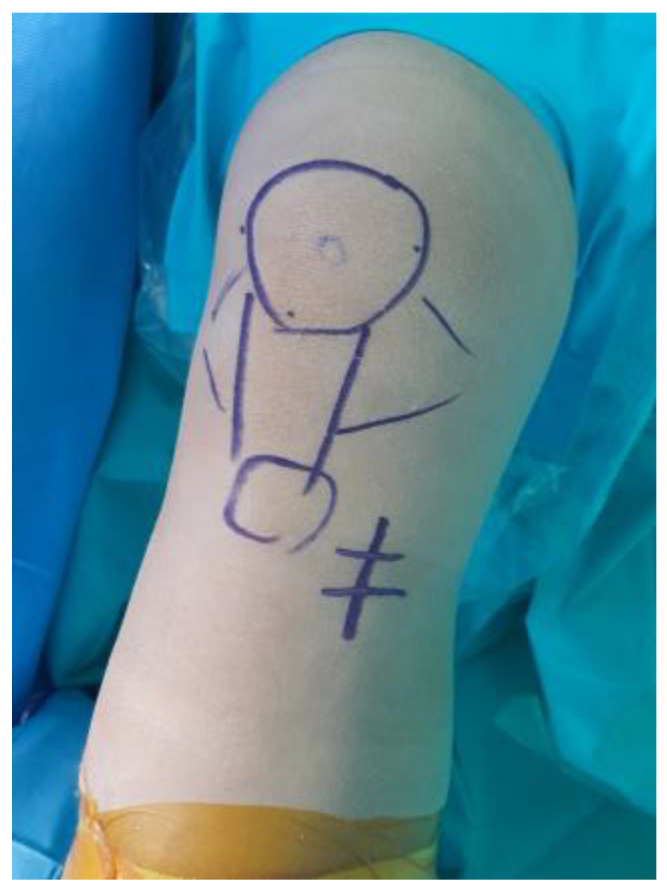
Vertical incision made at the anteromedial aspect of the proximal tibia.

**Figure 2 medicina-59-00922-f002:**
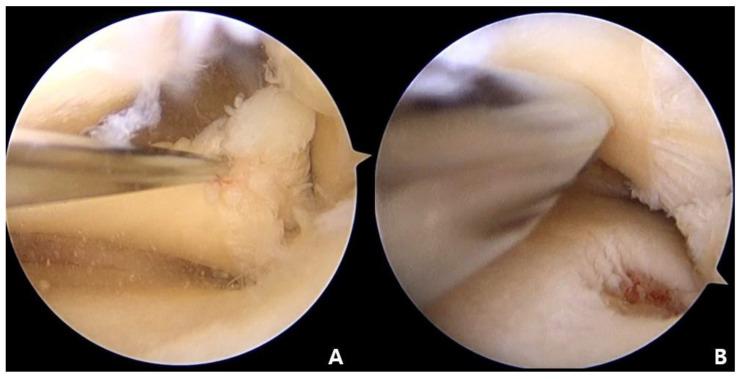
Arthroscopic findings: (**A**) arthroscopic view through the anterolateral portal of medial meniscus posterior horn in left knee; (**B**) bone bed decortication at the attachment site of the medial meniscus.

**Figure 3 medicina-59-00922-f003:**
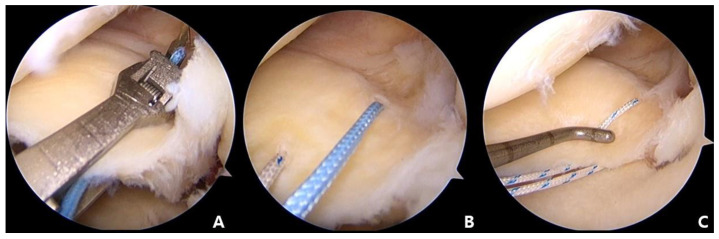
(**A**) The MMPH penetrated using a Knee Scorpion suture passer (Arthrex, Naples, FL, USA) at about 5 mm medial to the detached margin. (**B**) The second stitch located in the anterior position of the first stitch in the same manner. (**C**) Using the shuttle relay method, exchange of the first suture with the second suture to create a horizontal loop.

**Figure 4 medicina-59-00922-f004:**
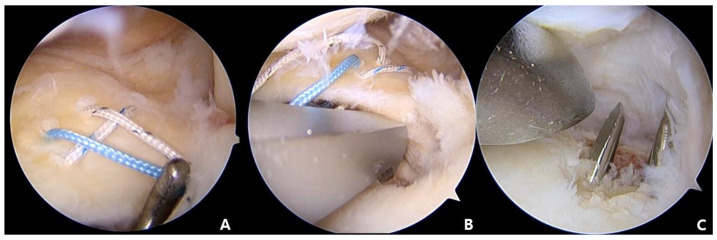
(**A**) Two vertical stitches overlaying and crossing the center of the horizontal suture. (**B**) Meniscus Root Repair System (Smith and Nephew, Andover, MA, USA) advanced using the AM portal, with the tip of the guide placed in the most medial side of the decorticated site of MMPR. (**C**) The second K-wire placed parallel and about 5 mm lateral to the first tunnel.

**Figure 5 medicina-59-00922-f005:**
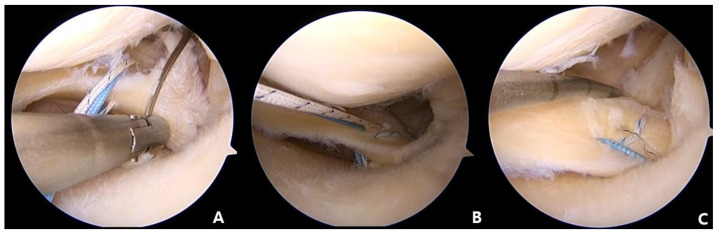
(**A**) Metal wire inserted into the tibial tunnel and pulled out through the AM portal using an arthroscopic grasper. (**B**) For the medial tunnel, passage of the two horizontal stitches and the two inferior vertical stitches, resulting in a total of 4 stitches. For the lateral tunnel, passage of the two superior vertical stitches. (**C**) An arthroscopic re-evaluation to confirm repair of the MMPH and to restore tension of the meniscus using arthroscopic probe device.

## Data Availability

Not applicable.

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
