# Peer review of "Posterior Medial Meniscus Root Repair Using Two Transtibial Tunnels with Modified Mason–Allen Stitches: A Technical Note"

_medicina, 2023, doi:10.3390/medicina59050922_

Round 1

Reviewer 1 Report

Dear Authors,

I was pleased to review the paper entitled " Posterior Medial Meniscus Root Repair using 2-Transtibial 2 Tunnels with Modified Mason-Allen stitches: A technical note" -

This article aims to describe a novel surgical technique using 2-transtibial tunnels with modified Mason-Allen stitches for Posterior Medial Meniscus Root Repair.

I read the article with interest, the title is well thought out and faithfully reflects the content of the study. 

Therefore, I found the content original, current, and relevant.

According to my opinion, some minor changes are needed to be considered suitable for publication:

The main limitation of the work is the lack of clinical data to support the procedure, even if only as case reports.

-The abstract is adequately developed and is useful to frame the characteristics and purpose of the study. 

-The introduction is quite complete but too short. 

I recommend adding information about the type of injury, (pathology risk factors, epidemiology, standard treatments performed etc..) supported by added bibliography.

Proper evaluation of the preoperative knee is critical to the success of the procedure, whether in arthroscopic or prosthetic pathology (MDPI: doi: 10.3390/medicina58091164)

-Materials and methods are absent. Support the surgical technique with clinical data. These sections are mandatory in MDPI instructions for author.

I recommend including the technical notes in materials and methods as in other papers already published in MDPI (https://doi.org/10.3390/jpm13010148).

Line 38 page 1, please insert image of “anteromedial aspect of the proximal tibia” incision.

Line 45 page 2, “MMPH” please clarify acronym if it is written first time.

Line 45 page 3, please explain “tie was performed under the periosteum with the knee at 30° flexion”

-The discussion and conclusion are quite well developed.

Line 126 “Use of the 2-tunnel technique may result in increased and even distribution of joint pressure between the MMPH and the bony bed” –  describe why you support this concept.

Line 127-1128 “Therefore, we thought that the 2-tunnel technique provides greater advantages than the 1-tunnel technique in biological healing. – explain which is the connection between biological healing and performing 1 or 2 transtibial tunnel

Author Response

Medicina reviewer’s comments

Reviewer 1

Dear Authors,

I was pleased to review the paper entitled " Posterior Medial Meniscus Root Repair using 2-Transtibial 2 Tunnels with Modified Mason-Allen stitches: A technical note" -

This article aims to describe a novel surgical technique using 2-transtibial tunnels with modified Mason-Allen stitches for Posterior Medial Meniscus Root Repair.

I read the article with interest, the title is well thought out and faithfully reflects the content of the study. 

Therefore, I found the content original, current, and relevant.

According to my opinion, some minor changes are needed to be considered suitable for publication:

The main limitation of the work is the lack of clinical data to support the procedure, even if only as case reports.

 è Thank you for your comment. I added clinical outcomes of included patients. Please see the “3. Case series”

-The abstract is adequately developed and is useful to frame the characteristics and purpose of the study. 

-The introduction is quite complete but too short. 

I recommend adding information about the type of injury, (pathology risk factors, epidemiology, standard treatments performed etc..) supported by added bibliography.

 è Thank you for your comment. We described additional background in detail.

Proper evaluation of the preoperative knee is critical to the success of the procedure, whether in arthroscopic or prosthetic pathology (MDPI: doi: 10.3390/medicina58091164)

 è Thank you for your comment. We additionally described “preoperative evaluations and indication”.

-Materials and methods are absent. Support the surgical technique with clinical data. These sections are mandatory in MDPI instructions for author.

I recommend including the technical notes in materials and methods as in other papers already published in MDPI (https://doi.org/10.3390/jpm13010148).

è We changed the title following to your recommendation.

Line 38 page 1, please insert image of “anteromedial aspect of the proximal tibia” incision.

è Thank you for your comment. We added the image of your comment. please see Fig 1.

Line 45 page 2, “MMPH” please clarify acronym if it is written first time.

è Thank you for your comment. We removed acronym and wrote full term.

Line 45 page 3, please explain “tie was performed under the periosteum with the knee at 30° flexion”

è Thank you for your comment. we modified that sentence more clearly.

-The discussion and conclusion are quite well developed.

Line 126 “Use of the 2-tunnel technique may result in increased and even distribution of joint pressure between the MMPH and the bony bed” –  describe why you support this concept.

è Thank you for your comment. we deleted that sentence. Instead, we described additional discussion about 1- or 2-tunnel technique.

Line 127-1128 “Therefore, we thought that the 2-tunnel technique provides greater advantages than the 1-tunnel technique in biological healing. – explain which is the connection between biological healing and performing 1 or 2 transtibial tunnel

è Thank you for your comment. we described additional discussion about 1- or 2-tunnel technique and added related references. Please see the same paragraph.

Reviewer 2 Report

Firstly, I appreciate the detailed description of the surgical technique used in the repair of medial meniscus posterior root tears.

The use of two transtibial tunnels with modified Mason-Allen stitches is an interesting approach that could provide improved outcomes in the repair of this type of injury.

However, I would like to suggest some areas for improvement. Firstly, the technical note would benefit from additional information on how patient selection is done, potential complications associated with the described technique.

The figures or images illustrating the surgical steps would are helpful in better understanding the procedure.

Overall, I believe that the technical note has potential to make a valuable contribution to the field.

I encourage you to consider these suggestions as you prepare your final manuscript.

Author Response

Firstly, I appreciate the detailed description of the surgical technique used in the repair of medial meniscus posterior root tears.

The use of two transtibial tunnels with modified Mason-Allen stitches is an interesting approach that could provide improved outcomes in the repair of this type of injury.

However, I would like to suggest some areas for improvement. Firstly, the technical note would benefit from additional information on how patient selection is done, potential complications associated with the described technique.

 è Thank you for your comment. We additionally described “preoperative evaluations and the indication”.

The figures or images illustrating the surgical steps would are helpful in better understanding the procedure.

è Thank you for your comment. We added the more image. According to reviewer 1, we inserted image of “anteromedial aspect of the proximal tibia” incision. please see Fig 1.

Overall, I believe that the technical note has potential to make a valuable contribution to the field.

I encourage you to consider these suggestions as you prepare your final manuscript.

Reviewer 3 Report

General suggestions.

I appreciate this is a brief note, and the article as written, is more or less a description of a novel surgical technical variation, without any validation (either biomechanically or clinically) that this technique is superior to the others. The authors acknowledge this in the last sentence of the conclusion, and to me it is unclear to what extent the current work qualifies for publication in this journal. I guess it is the decision of the editor whether a purely descriptive paper is of interest.

Specific suggestions.

INTRODUCTION

A complete tear of the medial meniscus posterior root (MMPR) can cause a complete 18 loss of hoop tension and increased contact pressure due to altered biomechanics of the 19 normal knee joint, which is like a complete meniscectomy or radial tear [1, 2]. Could you replace “which is like a” by “paralleling”…..

Numerous surgical methods have recently been used for the….. Maybe say “suggested” rather than “used”

as a result of the ability to bring back the tibiofemoral contact pressures and areas to the normal knee. Better say: “consequent to the idea that the contact pressure and contact are normalized”.

has been accepted as an excellent holding technique. Maybe stated “has been accepted as an effective technical approach”.

SURGICAL TECHNIQUE

Performance of a general arthroscopic examination using anterolateral (AL) and anteromedial (AM) portals is routine. Better say: “General arthroscopic…… was performed as routinely”.

surface of the tibial plateau, should then be identified arthroscopically after sMCL release. In the previous sentence, the authors state what WAS done, here they state what SHOULD be done. Please adopt a uniform style in this respect.

loaded with a No. 58 2 Ultrabraid (Smith and Nephew, Andover, USA) through the AM portal is then performed.  Thus far, the authors have used the past tense (as is correct for the method section), but now the present tense. Please adapt. This also concerns the following sentences, and I suggest you use past tense throughout.

K-wire was placed parallel and about 5 mm lateral side to the first tunnel. Please say “ 5 mm laterally” and omit the “side”.

For the medial tunnel, two horizontal stitches and two inferior vertical stitches are passed. Back to the present tense again, please stay in past tense throughout the method section.

DISCUSSION

Otherwise, repair of MMPRT can result in the 108 restoring of the hoop tension biomechanically. I am not entirely sure what the authors are trying to say, but I think the “otherwise” is not correctly used in that context.

The goal of 2-tunnel technique is recovering the normal anatomy….. . The word “recovering” should be “to recover”

in a narrow meniscus attach area…. I do not think this is proper English?

similar biomechanical results for the transtibial …. Please specify which test was done and which specific measure was compared.

may result in increased and even distribution of joint pressure…. First it should be “in AN increased”, but I am not sure how the terms “increased” and “even” fit together, maybe try to reformulate the sentence.

Therefore, we thought that the 2-tunnel technique provides greater advantages…..  I think the authors should state this differently and collect some evidence that the 2-tunnel technique provides advantages.

CONCLUSIONS

the ideal technique is not yet to be established. This should be “had not yet been established”

This relatively simple and effective procedure will be beneficial in restoration the function of the medial meniscus in patients who present MMPRT. Will it? Do the authors have collected any evidence that this is so? This should be part of the conclusion.

Author Response

Specific suggestions.

INTRODUCTION

A complete tear of the medial meniscus posterior root (MMPR) can cause a complete 18 loss of hoop tension and increased contact pressure due to altered biomechanics of the 19 normal knee joint, which is like a complete meniscectomy or radial tear [1, 2]. Could you replace “which is like a” by “paralleling”…..

è Thank you for your comment. We revised that sentence.

Numerous surgical methods have recently been used for the….. Maybe say “suggested” rather than “used”

è Thank you for your comment. We revised that sentence.

as a result of the ability to bring back the tibiofemoral contact pressures and areas to the normal knee. Better say: “consequent to the idea that the contact pressure and contact are normalized”.

è Thank you for your comment. We revised that sentence.

has been accepted as an excellent holding technique. Maybe stated “has been accepted as an effective technical approach”.

è Thank you for your comment. We revised that sentence.

SURGICAL TECHNIQUE

Performance of a general arthroscopic examination using anterolateral (AL) and anteromedial (AM) portals is routine. Better say: “General arthroscopic…… was performed as routinely”.

è Thank you for your comment. We revised that sentence.

surface of the tibial plateau, should then be identified arthroscopically after sMCL release. In the previous sentence, the authors state what WAS done, here they state what SHOULD be done. Please adopt a uniform style in this respect.

è Thank you for your comment. As per your comment, there are several unnecessary words in this sentence. So we revised that sentence to reduce confusion.

loaded with a No. 58 2 Ultrabraid (Smith and Nephew, Andover, USA) through the AM portal is then performed.  Thus far, the authors have used the past tense (as is correct for the method section), but now the present tense. Please adapt. This also concerns the following sentences, and I suggest you use past tense throughout.

è Thank you for your comment. As your comment, we modified sentence as past tense.

K-wire was placed parallel and about 5 mm lateral side to the first tunnel. Please say “ 5 mm laterally” and omit the “side”.

è Thank you for your comment. We modified that phrase.

For the medial tunnel, two horizontal stitches and two inferior vertical stitches are passed. Back to the present tense again, please stay in past tense throughout the method section.

è Thank you for your comment. As your comment, we modified sentence as past tense.

DISCUSSION

Otherwise, repair of MMPRT can result in the 108 restoring of the hoop tension biomechanically. I am not entirely sure what the authors are trying to say, but I think the “otherwise” is not correctly used in that context.

è Thank you for your comment. We changed the “otherwise” to “Instead”.

The goal of 2-tunnel technique is recovering the normal anatomy….. . The word “recovering” should be “to recover”

è Thank you for your comment. We modified that phrase.

in a narrow meniscus attach area…. I do not think this is proper English?

è Thank you for your comment. We modified that sentence more naturally.

similar biomechanical results for the transtibial …. Please specify which test was done and which specific measure was compared.

 è Thank you for your comment. we additionally described the detail of the reference.

may result in increased and even distribution of joint pressure…. First it should be “in AN increased”, but I am not sure how the terms “increased” and “even” fit together, maybe try to reformulate the sentence.

è Thank you for your comment. I agree with your opinion. So we removed that sentence and revised that paragraph according to review 1’s comment.

Therefore, we thought that the 2-tunnel technique provides greater advantages…..  I think the authors should state this differently and collect some evidence that the 2-tunnel technique provides advantages.

è Thank you for your comment. Reviewer 1 also pointed out that sentence. So we described additional discussion about 1- or 2-tunnel technique and added related references. Please see the same paragraph.

CONCLUSIONS

the ideal technique is not yet to be established. This should be “had not yet been established”

è Thank you for your comment. We modified that sentence.

This relatively simple and effective procedure will be beneficial in restoration the function of the medial meniscus in patients who present MMPRT. Will it? Do the authors have collected any evidence that this is so? This should be part of the conclusion.

è Thank you for your comment. As for your comment, we modified that sentence and described the necessity of the well-designed study to support our conclusion.

Reviewer 4 Report

Dear Authors,

Thank you for sending this innovative surgical technique.

But I have a few serious concerns about this study.

1. Please explain the biomechanical advantage of two tibial tunnels over one tibial tunnel? Though the authors have listed some advantages over the conventional technique, Has any biomechanical study been done on this issue?

2. Please list the number of cases that have been attempted by this technique . There is no mention of the number of cases in the technical note

3. What is the average follow-up done?

4. Has any functional outcome measurement been done? Please provide the results of the functional outcome measurement.

Thank you.

Author Response

Dear Authors,

Thank you for sending this innovative surgical technique.

But I have a few serious concerns about this study.

  1. Please explain the biomechanical advantage of two tibial tunnels over one tibial tunnel? Though the authors have listed some advantages over the conventional technique, Has any biomechanical study been done on this issue?

è Thank you for your comment. As your comment, there are few studies related 1 or 2 tunnel techniques.  we described additional discussion about 1- or 2-tunnel technique and added related references. Please see the same paragraph (line 155-168).

  1. Please list the number of cases that have been attempted by this technique . There is no mention of the number of cases in the technical note

 è Thank you for your comment. I added clinical outcomes of included patients. Please see the “3. Case series”

  1. What is the average follow-up done?

 è Thank you for your comment. I added clinical outcomes of included patients. Please see the “3. Case series”

  1. Has any functional outcome measurement been done? Please provide the results of the functional outcome measurement.

  è Thank you for your comment. I added clinical outcomes of included patients. Please see the “3. Case series”

Thank you.

Round 2

Reviewer 1 Report

The Author correctly responded to the review

Reviewer 4 Report

thank you for your good work.